# Cover crop and crop residue removal effects on temporal dynamics of soil carbon and nitrogen in a temperate, humid climate

**Inderjot Chahal**, **Laura L. Van Eerd***

School of Environmental Sciences, University of Guelph, Ridgetown, Ontario, Canada

* lvaneerd@uoguelph.ca

**Data Availability Statement:** All relevant data are within the paper and its Supporting Information files.

## Abstract

Quantification of seasonal dynamics of soil C and N pools is crucial to understand the land management practices for enhancing agricultural sustainability. In a cover crop (CC) experiment established in 2007 and repeated at an adjacent site in 2008, we evaluated the medium-term impact of CC (no cover crop control (no-CC), oat (*Avena sativa* L.), oilseed radish (OSR, *Raphanus sativus* L. var. *oleoferus* Metzg. Stokes), winter cereal rye (rye, *Secale cereale* L.), and a mixture of OSR+Rye) and crop residue management (residue removed (-R) and residue retained (+R)) on soil C and N dynamics and sequestration. Labile and stable fractions of C and N were determined at seven different time points from 0–15 cm depth during tomato (*Solanum lycopersicum* L.) growing season in 2015 and 2016 (referred to as site-years). As expected, over the tomato growing season in both site-years, organic C (OC) and total N did not change while the labile C and N fractions changed with greater concentrations observed at 2 weeks after tillage (WAT) and greater treatment differences observed for seven out of eleven soil attributes at tomato harvest. Therefore, 2WAT (early June) and tomato harvest (early September) are reasonably optimum sampling times for soil C and N attributes. Seasonal variation of labile fractions suggested the potential impact of substrate availability from crop residues on soil C and N cycling. Medium-term CC usage enhanced the surface soil C and N storage. Overall, this study highlights the positive and synergistic influences of CCs and maintaining crop residues in increasing both labile and stable fractions of C and N and enhancing soil quality in a temperate humid climate.

## Introduction

Winter wheat (*Triticum aestivum* L.) followed by processing tomato (*Solanum lycopersicum* L.) is a common vegetable crop rotation in Ontario, Canada, and Midwest USA. Residue management of winter wheat is important as it may influence the microbial processes, N availability, and C dynamics in the following season crop. Winter wheat straw has a high C:N (80:1), thus, leaving winter wheat straw in the field may cause N immobilization with a potential for crop N deficiency. Moreover, soils with crop residue left on the surface are colder and wetter which results in a delayed spring planting. Therefore, to avoid this, growers tend to remove

**Funding:** Authors are grateful to Ontario Ministry of Agriculture, Food, and Rural Affairs (OMAFRA), Grain Farmers of Ontario (GFO), and Ontario Processing Vegetable Growers (OPVG) for providing research funding to LVE. The funders had no role in study design, data collection and analysis, decision to publish, or preparation of the manuscript.

**Competing interests:** The authors have declared that no competing interests exist.

**Abbreviations:** CC, cover crop; $Cmin_{2d}$, cumulative 2d soil C mineralization; MBC, microbial biomass C; MBN, microbial biomass N; no-CC, no cover crop control; OSR, oilseed radish; OSR+Rye, mixture of oilseed radish and winter cereal rye; SLAN, Solvita labile amino N; OC, organic C; WAT, weeks after tillage; WAS, wet aggregate stability; WEOC, water extractable organic C; WEON, water extractable organic N; -R, crop residue removed; +R, crop residue retained..

winter wheat straw from the field after grain harvest. Crop residue, however, is a source of C and N for soil macro and micro fauna. Crop residue removal (-R) may have some negative implications on long-term soil productivity and quality [1,2] by reducing soil organic C (OC) and total N levels [3,4], reducing soil microbial activity [5], and increasing wind and water erosion leading to loss of nutrients [2]. However, inclusion of cover crops (CC) after crop residue removal may be an effective management strategy to offset the expected reductions in soil C inputs and soil quality [2].

Several benefits to agroecosystems have been observed with adoption of CC in crop rotations [6–9]. Improvements in soil physical [10–12], biological [2], and chemical [13,14] soil quality indicators are observed with CCs, which are highly dependent on the quantity and quality of CC biomass input to the soil [15]. Cover crops also maintain soil C and N levels by sequestering C and N from atmosphere and mitigating C and N losses [16]; thus, might increase N availability to the subsequent crop [16]. However, timing of N release from the CC residues is a critical factor, which is dependent on the CC C:N, lignin and cellulose concentrations of CC [17,18], and timing of CC termination and residue incorporation [19]. Additionally, soil moisture and temperature play an important role in influencing the soil processes involved in residue decomposition and nutrient cycling; thus, CC and crop residue effects on soil C and N dynamics is dependent on agroecosystem management and environmental factors.

Soil OC and total N change slowly over the long-term and are considered as the indicators of stable fraction of soil C and N, whereas SLAN (Solvita labile amino N), Solvita $CO_2$-burst, microbial biomass C and N (MBC and MBN), water extractable organic C and N (WEOC and WEON), wet aggregate stability (WAS), total inorganic N, and cumulative 2d C mineralization ($Cmin_{2d}$), are known to vary over short (seasonal) term, and thus, are the indicators of labile fraction of C and N. Seasonal variability (across the growing season) in labile fractions of C and N [20–24] is primarily dependent on the quantity of crop residue produced, rhizodeposition during crop growth, and soil temperature and precipitation, which influence the soil microbial activity and residue decomposition [25]. Microbial activity is a key factor affecting the cycling of labile pools of C [26] and N. For instance, MBC and MBN respond quickly to the addition/incorporation of crop residue in the soil. Moebius et al. [21] reported significant temporal variation in soil physical indicators, such as WAS, over a crop growing season in a tillage system. Similarly, soil respiration, water extractable fractions are very sensitive and responsive to land management practices [27,28]. Therefore, soil indicators of labile fractions of C and N are very useful for detecting the initial changes in the status of soil organic matter which affects the nutrient turnover [29]. Moreover, evaluation of short-term changes in the soil C and N fractions, especially after CCs, provides valuable information on microbial biomass dynamics, substrate availability, timing of N mineralization, and the net balance between N mineralization and immobilization. A better understanding of the seasonal dynamics of soil C and N will advance our knowledge for improving agricultural management practices focusing on enhancing agricultural sustainability, soil quality, and climate change mitigation. Thus, assessments of seasonal variability in labile fractions of soil C and N are needed.

The inclusion of CCs in crop rotations may provide greater rotational and temporal diversity in the production systems [30]; thus, resulting in an increase in microbial activity [31] cycling of C and N, and decomposition of biomass and C assimilation [32]. Furthermore, the temporal variety in the plant inputs (observed with CC in the crop rotations) and residue quantity and quality have a strong influence on soil functions and processes, which may have implications on increasing crop yields [30]. Adoption of CCs also result in an increase in the soil OC which plays an important role in enhancing the soil functionality, processes, and sustainability.

To date, there is a limited knowledge on the potential interactive effects of CCs and crop residue removal on soil C and N dynamics. Additionally, CC and residue removal impacts on soil C and N accumulation are expected to be observed in the medium- to long-term as opposed to the short-term (see reviews by [2,3,11]). Therefore, the medium-term CC experiment used in this study was ideally suited for evaluating the CC- and crop residue-induced effects on C and N dynamics. The main objective of the study was to assess the mechanism of C and N accumulation with CCs and crop residue management (removed (-R) and retained (+R)). We hypothesize that a potential synergistic effect of enhancing crop diversity by including CCs and maintaining crop residues will be observed on soil C and N sequestration. The study is expected to advance our understanding of the integrative effect of CCs and crop residue removal on soil processes involved in C and N cycling.

## Materials and methods

### Site description and experimental design

The medium-term CC experiment, established in 2007 and repeated an adjacent site in 2008 at Ridgetown, Ontario, Canada, was used for studying soil C and N dynamics. Site-year was used to clearly indicate that soil and crop sampling was conducted from different sites in different years. Soil characteristics are described in Table 1. Since the initiation of the CC experiment in 2007 and 2008, crop rotation consisted of grain and processing vegetable crops, typical of southwestern Ontario (Table 2). As described previously [6,33–36], experimental design consisted of a split-plot arranged as a randomized complete block with four replicates. Summer-planted (July, August, September) annual CC treatments were grown in the main plots since 2007 and 2008. In addition to a no-CC, four CC treatments were evaluated which were oat (*Avena sativa* L.), oilseed radish (OSR) (*Raphanus sativus* L. var. *oleoferus* Metzg. Stokes), winter cereal rye (rye, *Secale cereale* L.), and a mixture of OSR+Rye. Cover crops were not planted after grain corn harvest in late October/early November due to the cold temperatures; after soybean harvest, winter wheat was planted instead of CCs. Crop residue management was applied in the split-plots three times since the experiment started at both sites (in 2009 and 2010 after spring wheat harvest, 2011 and 2012 after grain corn harvest, and 2014 and 2015 after winter wheat harvest) in the production system. Thus, from 2007 to 2015 and 2008 to 2016, CCs were planted six times in the crop rotation (Table 2). Cover crop attributes

**Table 1. Select soil[z] properties from 0–15 cm depth at Ridgetown, Ontario, Canada during 2015 and 2016.**

| Property[y] | Site-year 2015 | Site-year 2016 |
|---|---|---|
| Soil texture | Sandy loam (Orthic humic gleysol) | |
| Particle size distribution | | |
| Sand (%) | 76.7±0.26 | 76.3±0.31 |
| Silt (%) | 16.9±0.24 | 18.5±0.29 |
| Clay (%) | 6.34±0.14 | 5.07±0.20 |
| pH | 6.12±0.05 | 7.06±0.04 |
| Phosphorus (mg kg$^{-1}$) | 5.60±0.17 | 21.4±1.55 |
| Potassium (mg kg$^{-1}$) | 136±3.93 | 147±5.87 |

[z] Means of a composite sample (sixty soil cores of 3.5 cm diameter) taken from each cover crop and crop residue treatment plot (n = 40) at tomato harvest in September 2015 and 2016.

[y]Methods included particle size via hydrometer, pH was 1:1 (soil:water) by volume, P was Olsen bicarbonate extraction and K was ammonium acetate extraction.

**Table 2. In a medium-term cover crop experiment, crop rotation and crop residue management from 2007 to 2015 (site-year 2015) and 2008 to 2016 (site-year 2016).**

| Site-year 2015 | Site-year 2016 | Main crop | Fall planting |
|---|---|---|---|
| 2007 | 2008 | Processing pea | Cover crops |
| 2008 | 2009 | Processing sweet corn | Cover crops |
| 2009 | 2010 | Spring wheat | Cover crops |
| 2010 | 2011 | Processing tomato | Cover crops |
| 2011 | 2012 | Grain corn -stover removal | None |
| 2012 | 2013 | Processing squash | Cover crops |
| 2013 | 2014 | Soybean | Winter wheat |
| 2014 | 2015 | Winter wheat -straw removal | Cover crops |
| 2015 | 2016 | Processing tomato | Cover crops |

(biomass, C and N concentration) were quantified three times (late October/early November, April, and May) in 2014–15 and 2015–16.

All the crop management and field operations have been previously described in Chahal and Van Eerd [34]. Two weeks prior to tomato transplanting, P and K fertilizers were applied at the rate of 94.1 kg ha$^{-1}$ ($P_2O_5$) and 123 kg ha$^{-1}$ ($K_2O$) [33]. Soil was tilled on May 19, 2015 and May 20, 2016 to incorporate CC residues and fertilizers and to prepare for tomato transplanting. At tomato transplanting, a starter N fertilizer (15 kg N ha$^{-1}$) was applied with water [6,33]. No irrigation was applied to tomato crop.

## Soil sampling and analysis

In both site-years, soil was sampled from 0–15 cm depth at seven different times during tomato growing season. Sampling regime was based on tillage; 3 weeks before tillage on April 29, 2015 and April 27, 2016; at tillage on May 19, 2015 and May 20, 2016; and two, four, eight and twelve weeks after tillage (WAT) on June 5, June 16, June 29, July 29 in both site-years; and at tomato harvest on September 11, 2015 and September 6, 2016. A composite sample of random 20 soil cores of 3.5 cm diameter was taken from each split-plot and soil samples were homogenized by hand in the field.

Sample handling and sieving for OC [37], total N [38], $Cmin_{2d}$ [39], MBC and MBN [40] was similar; soil was sieved through 4 mm mesh screen and protocols described in Carter and Gregorich [41] were followed. Until processing, soil samples for OC, total N, $Cmin_{2d}$, and total inorganic N (TIN) were stored in a cooler (4°C); MBC and MBN were stored in the freezer (-20°C). A sub-sample of soil (5 g for OC and total N, and 80 g for $Cmin_{2d}$) was oven-dried at 50°C for 24 hours. A ground (mortar and pestle) sub-sample of oven-dried soil (0.15 g) was used for quantifying OC and total N on LECO CN analyzer (Leco Corporation, St. Joseph, MI). For evaluating $Cmin_{2d}$, substrate-induced respiration was conducted with an addition of glucose (1.6 mg C g$^{-1}$ dry soil) using an alkali trap method with 60 g dry soil [35,42]. Soil organic C and $Cmin_{2d}$ were expressed as mg C g$^{-1}$ dry soil; total N and TIN were expressed as mg N g$^{-1}$ dry soil. Microbial biomass C and N were measured by using chloroform-extraction method as described by Carter and Gregorich [41] and Vance et al. [43]. Fifteen grams of moist soil sub-sample was fumigated, in a sealed glass desiccator kept under vacuum for 24 hrs in the dark in a fume hood, using 50 mL of ethanol-free chloroform which was placed in a beaker along with boiling chips; whereas another 15 g soil was non-fumigated. Both fumigated and non-fumigated soil samples were extracted using 30 mL of 0.5 M $K_2SO_4$ solution, and filtrates were analyzed for C and N via dry combustion on a LECO CN analyzer (Leco Corporation, St. Joseph, MI). Microbial biomass C and MBN were quantified as the difference of

fumigated and non-fumigated samples and were expressed as µg C g$^{-1}$ and µg N g$^{-1}$, respectively. The $k_{EC}$ and $k_{EN}$ coefficients used for MBC and MBN were 0.45 and 0.18, respectively [41,44]. Wet aggregate stability was measured by using 4 g un-sieved air-dried soil in an apparatus (Eijelkamp Agrisearch Equipment 08.13, Giesbeek, The Netherlands) similar to Kemper and Rosenau [45] and described in detail by Van Eerd et al. 2018 [46].

For Solvita $CO_2$-burst, water extractable organic C (WEOC), water extractable organic N (WEON), and Solvita labile amino N (SLAN), soil was sieved through 2 mm mesh screen and 60 g soil sub-sample was oven-dried at 40˚C for 24 hrs [33,47–50]. Briefly, 4 g of oven-dried soil was used for quantifying SLAN, WEOC, and WEON; Solvita $CO_2$-burst was measured using 40 g dry soil (see Chahal and Van Eerd [33] for details). For quantifying WEOC and WEON, 4 g oven-dried soil was extracted with 40 mL distilled water by mechanically shaking on an orbital shaker for 10 mins, followed by filtration and analyzing the extracts for inorganic N (on an auto analyzer SEAL Analytics) and total and inorganic C (on LECO CN analyzer) [33]. Soil with the gel paddles was incubated for 24 hrs in a sealed glass beaker to evaluate Solvita $CO_2$-burst and SLAN. After 24 hr incubation, gel paddles changed the colour due to evolved $CO_2$ (Solvita) and $NH_3$ (SLAN); colour intensity measurements were taken using a digital colorimeter reader [33]. Concentrations of SLAN, WEON, WEOC, and Solvita $CO_2$-burst were expressed on dry weight basis as mg $NH_3$-N kg$^{-1}$, mg N kg$^{-1}$, mg C kg$^{-1}$, and mg $CO_2$-C kg$^{-1}$.

### Data analysis

All fractions of C and N were expressed on a soil dry weight concentration basis. Soil OC and total N represented the stable fractions of soil C and N, whereas all other evaluated parameters indicated the labile pools of C and N. Both site-years were evaluated separately as initial statistical analysis detected an interaction between site-year and treatments. In each site-year, fixed effects of CC, crop residue management, and sampling time, and their interactions on soil attributes were assessed using repeated measures analysis with PROC GLIMMIX in SAS (SAS Institute, version 9.4 Cary, NC, USA). Replication, and replication by CC were the random effects [51]. To account for the repeated measure analysis, a random statement of replication by sampling time was included. Analysis of residuals and normality test (Shapiro-Wilk) was conducted to confirm the assumptions of ANOVA [52]. Based on the output of studentized conditional residuals, no outliers were detected. All assumptions of ANOVA were met, therefore, no adjustments to covariance structure and data transformations were conducted. A protected LSD test was used to compare treatment means at α = 0.05. Additionally, radar charts were prepared for both crop residue treatments at tomato harvest in each site-year; where data were expressed as a ratio of the average indicator value for each CC treatment and the maximum value of the respective attribute, to facilitate the visual interpretation of the impact of management on soil parameters. As mentioned previously, microbial activity is a critical function governing soil organic matter formation [30]; therefore, Pearson correlations of OC-related processes (MBC, MBN, and WAS) with labile fractions (Cmin$_{2d}$, WEOC, WEON, TIN, Solvita $CO_2$-burst and SLAN) stable fractions (OC, total N) were assessed.

### Tomato growing season

Air temperature and precipitation during the tomato growing season in site-years 2015 and 2016 were collected by an Environment Canada weather station located < 1 km from the experiment (Fig 1). Spring 2016 (April and May) was drier than spring 2015 by 69 mm (Fig 1) and 30-yr mean by 52.8%, whereas spring 2015 had a relatively similar total precipitation (74.7 mm) as 30-yr mean (76 mm). Average monthly air temperature during most of the tomato growing season from June to August was comparable in both site-years (mean 24.7˚C in site-

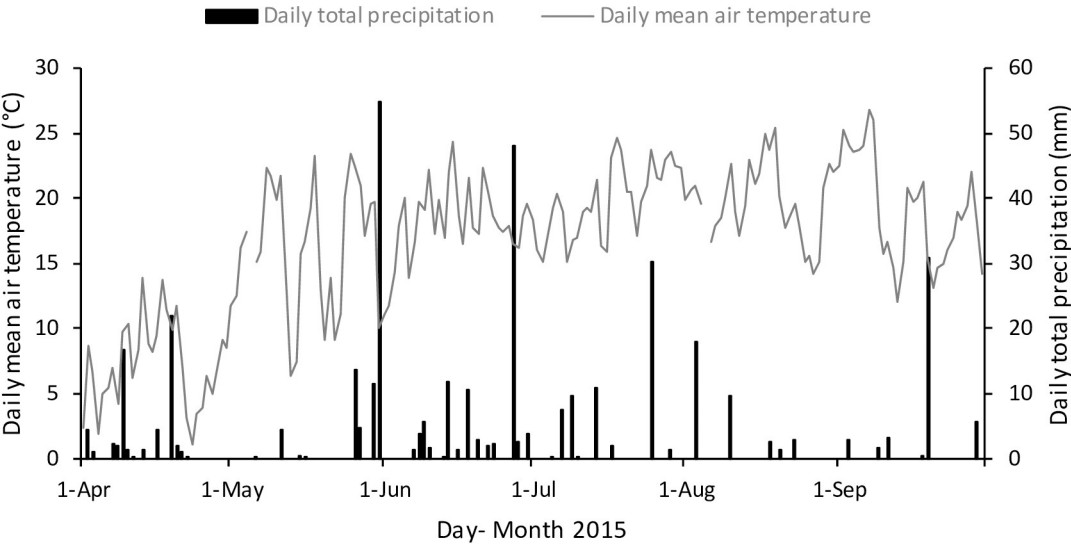

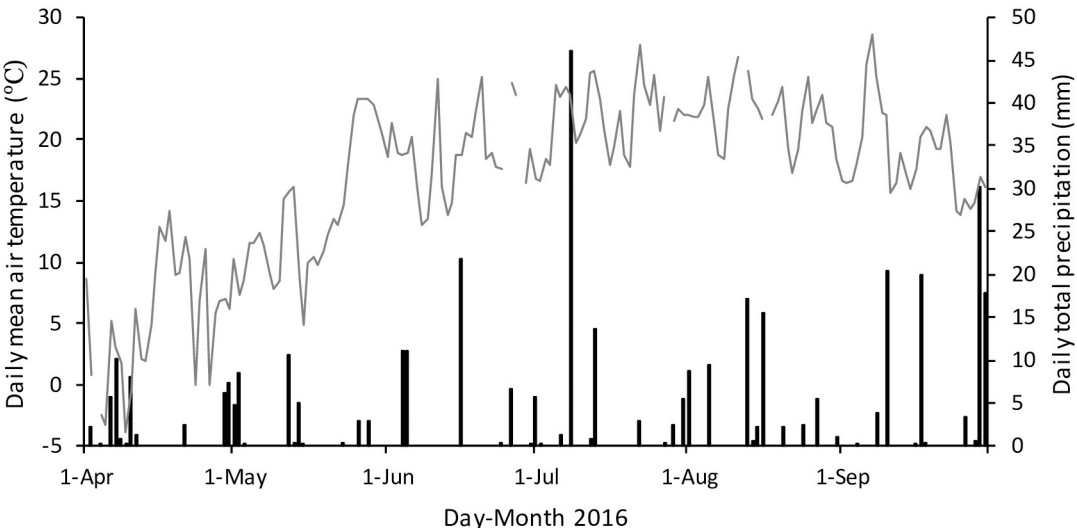

**Fig 1. Daily total precipitation (mm) and mean air temperature (˚C) during the tomato growing season in 2015 and 2016 at Ridgetown, Ontario, Canada.** Dash lines with arrows represent the soil sampling dates in site-year 2015 and 2016.

year 2015; mean 25.8˚C in site-year 2016) but warmer than 30-yr mean (21.4˚C). In contrast to air temperature, total monthly precipitation from June to August in sampling years was lower than 30-yr mean by 20.3 and 19.3 mm, respectively. However, total precipitation during tomato growing season in both years was equivalent (65.7 mm in 2015 and 65.5 mm in 2016; Fig 1). Regarding the temporal distribution of precipitation relative to soil sampling, June and July had the greatest total precipitation in site-year 2015, while early July was greatest in site-year 2016 (Fig 1). Despite differences in weather between years, in both site-years there were no effects of CC and crop residue management treatment on soil gravimetric moisture content (S1 Table). In site-year 2015, soil moisture content in April was greater than June, July, and September. No seasonal variability in soil gravimetric moisture content was detected in site-year 2016.

## Results

### Cover crop attributes

In fall of both site-years, OSR had the greatest CC biomass, C and N content (avg. 4850 kg ha$^{-1}$, 1780 kg C ha$^{-1}$, 159 kg N ha$^{-1}$, respectively) whereas cereal rye had the lowest (1960 kg ha$^{-1}$, 824 kg C ha$^{-1}$, and 64 kg N ha$^{-1}$, respectively; Fig 2). Over 6-yrs from fall 2007 to fall 2014, average annual CC biomass across all CC treatments was 3220±115 kg ha$^{-1}$, C was 1280±46.2 kg C ha$^{-1}$, and N was 78±3.47 kg N ha$^{-1}$ (Fig 2). Similarly, from fall 2008 to fall 2015, average annual CC biomass across all CC treatments was 3520±116 kg ha$^{-1}$, C was 1290±50.7 kg C ha$^{-1}$, and N was 97.7±4.15 kg N ha$^{-1}$ (Fig 2). Cover crop C concentration was not different among CC treatments in both springs (2015 and 2016, S2 Table). In spring 2015, C:N of CC was 18.4–32.3 with the trend of OSR≥OSR+Rye = rye≥oat (S2 Table). In contrast, C:N of CC in spring (avg. 19.2±1.54) in site-year 2016 was not different among CC treatments (S2 Table).

### Soil attributes

In both site-years, concentration of total N did not change over the tomato growing season (Tables 3 and 4). However, a significant CC by time interaction for OC was detected in both site-years (Tables 3 and 4) which was attributed to the lowest OC concentration in no-CC plots at tomato harvest. In both site-years, CC treatment differences, but no two-way interaction of CC by time, were detected on total N concentration with no-CC being the lowest (S3 and S4 Tables). In addition to medium-term CC effects on OC and total N, the +R treatment had greater OC concentration than -R in site-year 2015 ($P < 0.0001$) but no effect in site-year 2016 ($P = 0.9888$).

In contrast to stable fractions, labile C and N fractions varied temporally with CC and crop residue management. Seasonal variability with CCs was detected in all labile fractions in site-year 2015 (Table 3), but in site-year 2016, 6 out of 9 fractions changed temporally with CCs (Table 3). Moreover, compared with the CCs, interaction between crop residue management and time was detected in only two labile fractions (SLAN and $Cmin_{2d}$ in site-year 2015, SLAN and total inorganic N in site-year 2016; Tables 3 and 4). Across tomato growing season, no-CC had the lowest $Cmin_{2d}$, MBC, and MBN concentrations in site-year 2015 (Fig 3), and the lowest MBC and Solvita $CO_2$-burst concentration in site-year 2016 (Fig 4). The CC by time interaction in site-year 2015 for Solvita $CO_2$-burst, MBC, MBN, and $Cmin_{2d}$ was due to lowest concentrations in no-CC (oat intermediate for Solvita, $Cmin_{2d}$) in April but these differences were minimized at other time points (Fig 3). A sharp decrease was observed in the no-CC treatment only for MBC concentration from April to 2WAT followed by a levelling in the values until tomato harvest in site-year 2016 (Fig 4). For WAS, greater CC treatment differences were detected at April in site-year 2016 (Fig 4). As expected, WEON was highly variable in its temporal response to CCs and hence, no clear trend was detected in CC treatments across different sampling times (Fig 3). Larger pools of labile C and N with CC than no-CC across the season confirm the positive influences of CC on C and N dynamics in our medium-term experiment.

Retaining crop residues in the field resulted in greater concentrations of WAS, $Cmin_{2d}$, Solvita $CO_2$-burst, and total N in site-year 2015, and WAS, total inorganic N, and WEOC in site-year 2016 than removing residues (S3 and S4 Tables). Over the season, +R had greater concentrations of labile fractions ($Cmin_{2d}$ in site-year 2015 and total inorganic N in 2016) than -R. For $Cmin_{2d}$, interaction between crop residue treatments and time was due to greater values in +R than -R at all sampling times, except tomato harvest. In contrast, total inorganic N concentration in April was not different between +R and -R, while +R had greater concentration than

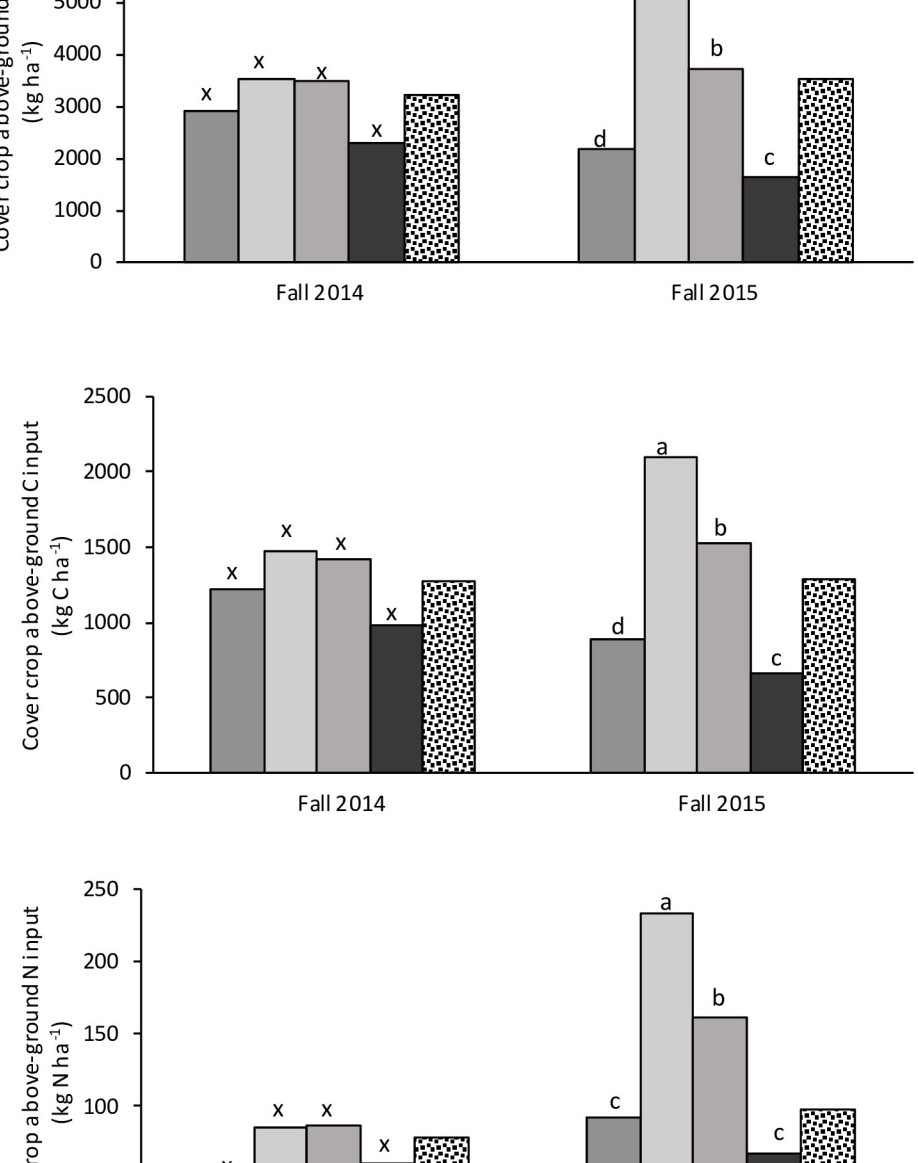

**Fig 2. In a medium-term cover crop (CC) experiment, annual above-ground CC biomass, C and N inputs in fall 2014, fall 2015, and average over a period of 6-yrs from various CC treatments.** For each parameter in each year, bars followed by a different letter indicate statistical significance per protected LSD test (*P*<0.05; n = 8).

-R across all other sampling times; thus, leading to an interaction between sampling time and crop residue treatments (data not shown). Therefore, these results of larger C and N pools of both labile and stable fractions with +R than -R further indicates the potential of +R in increasing soil fertility, functionality, and C and N cycling.

**Table 3. Probability values for the main effects of medium-term summer-planted cover crop (6-yrs), crop residue management, sampling time, and interactions on soil stable and labile fractions of C and N sampled from 0–15 cm depth in site-year 2015.**

| | | Cover crop (CC) | Crop residue (R) | Time (T) | CC x R | CC x T | R x T | CC x R x T |
|---|---|---|---|---|---|---|---|---|
| Fraction | Unit | —————————————————————————————Pr > F—————————————————————————————— | | | | | | |
| OC | mg C g$^{-1}$ | <0.0001 | 0.0538 | 0.1969 | 0.1708 | <0.0001 | <0.0001 | 0.0903 |
| Total N | mg N g$^{-1}$ | <0.0001 | <0.0001 | 0.1337 | 0.0782 | 0.2797 | 0.4344 | 0.8157 |
| WAS | % | <0.0001 | <0.0001 | <0.0001 | <0.0001 | <0.0001 | 0.3619 | 0.0126 |
| Cmin$_{2d}$ | mg C g$^{-1}$ | <0.0001 | <0.0001 | <0.0001 | 0.5802 | <0.0001 | 0.0002 | 0.9904 |
| Solvita | mg CO$_2$-C kg$^{-1}$ | 0.0138 | 0.0011 | <0.0001 | 0.6759 | <0.0001 | 0.7229 | 0.9986 |
| SLAN | mg NH$_3$-N kg$^{-1}$ | <0.0001 | 0.7781 | <0.0001 | 0.0038 | <0.0001 | <0.0001 | 0.0460 |
| WEOC | mg C kg$^{-1}$ | <0.0001 | 0.0027 | 0.6753 | <0.0001 | 0.0037 | 0.0720 | 0.0015 |
| WEON | mg N kg$^{-1}$ | 0.1496 | 0.0587 | 0.1765 | 0.0511 | 0.0189 | 0.1469 | 0.1316 |
| Total inorganic N | mg N g$^{-1}$ | <0.0001 | 0.0048 | 0.2873 | <0.0001 | 0.0059 | 0.2318 | 0.0027 |
| MBC | µg C g$^{-1}$ | <0.0001 | 0.7000 | <0.0001 | 0.7589 | 0.0159 | 0.7386 | 0.9591 |
| MBN | µg N g$^{-1}$ | <0.0001 | 0.7830 | <0.0001 | 0.9213 | 0.0302 | 0.7369 | 0.9205 |

Bold font indicates statistical significance at *P*<0.05.

Cmin$_{2d}$, cumulative 2d soil carbon mineralization; MBC, microbial biomass C; MBN, microbial biomass N; SLAN, Solvita labile amino N; OC, soil organic C; WAS, wet aggregate stability; WEOC, water extractable organic C; WEON, water extractable organic N.

In addition to the two-way interactions described above, three-way interactions between CC, crop residue treatment, and sampling time (Tables 3 and 4) were detected on SLAN in both site-years, and WAS, WEOC, and TIN in site-year 2015 (Fig 5). In both site-years, no-CC +R and no-CC-R had the greatest SLAN concentration at all time points (Fig 5). In site-year 2015 only, the least differences in SLAN concentrations among treatments were observed at harvest (Fig 5). Least treatment differences for WEOC and total inorganic N were detected in mid-June to July (Fig 5). As expected, WEOC and total inorganic N were highly variable temporally; no clear trend was detected in CC and crop removal treatments, however, cereal rye had the least concentration over the season (Fig 5). Therefore, these results of variable response

**Table 4. Probability values for the main effects of medium-term summer-planted cover crop (6-yrs), crop residue management, sampling time, and interactions on soil stable and labile fractions of C and N sampled from 0–15 cm depth in site-year 2016.**

| | | Cover crop (CC) | Crop residue (R) | Time (T) | CC x R | CC x T | R x T | CC x R x T |
|---|---|---|---|---|---|---|---|---|
| Fraction | Unit | —————————————————————————————Pr > F—————————————————————————————— | | | | | | |
| OC | mg C g$^{-1}$ | 0.0003 | 0.8278 | 0.1055 | 0.4242 | <0.0001 | 0.9888 | 0.7518 |
| Total N | mg N g$^{-1}$ | <0.0001 | 0.7065 | 0.0536 | 0.5112 | 0.4459 | 0.4296 | 0.4355 |
| WAS | % | 0.0122 | <0.0001 | 0.0002 | 0.7240 | <0.0001 | 0.6911 | 0.9501 |
| Cmin$_{2d}$ | mg C g$^{-1}$ | 0.6454 | 0.2510 | <0.0001 | 0.0581 | 0.1546 | 0.8764 | 0.9965 |
| Solvita | mg CO$_2$-C kg$^{-1}$ | <0.0001 | 0.6009 | <0.0001 | 0.7403 | <0.0001 | 0.0985 | 0.8098 |
| SLAN | mg NH$_3$-N kg$^{-1}$ | <0.0001 | 0.5351 | <0.0001 | <0.0001 | <0.0001 | 0.0119 | <0.0001 |
| WEOC | mg C kg$^{-1}$ | <0.0001 | <0.0001 | <0.0001 | <0.0001 | 0.0009 | 0.2326 | 0.4115 |
| WEON | mg N kg$^{-1}$ | 0.0708 | 0.2796 | <0.0001 | 0.1069 | 0.7595 | 0.4980 | 0.3860 |
| Total inorganic N | mg N g$^{-1}$ | 0.0001 | <0.0001 | <0.0001 | <0.0001 | 0.0254 | <0.0001 | 0.8429 |
| MBC | µg C g$^{-1}$ | <0.0001 | 0.8096 | 0.1674 | 0.5194 | 0.0254 | 0.6503 | 0.6044 |
| MBN | µg N g$^{-1}$ | <0.0001 | 0.2688 | 0.0892 | 0.1489 | 0.2379 | 0.5648 | 0.1883 |

Bold font indicates statistical significance at *P*<0.05.

Cmin$_{2d}$, cumulative 2d soil carbon mineralization; MBC, microbial biomass C; MBN, microbial biomass N; SLAN, Solvita labile amino N; OC, soil organic C; WAS, wet aggregate stability; WEOC, water extractable organic C; WEON, water extractable organic N.

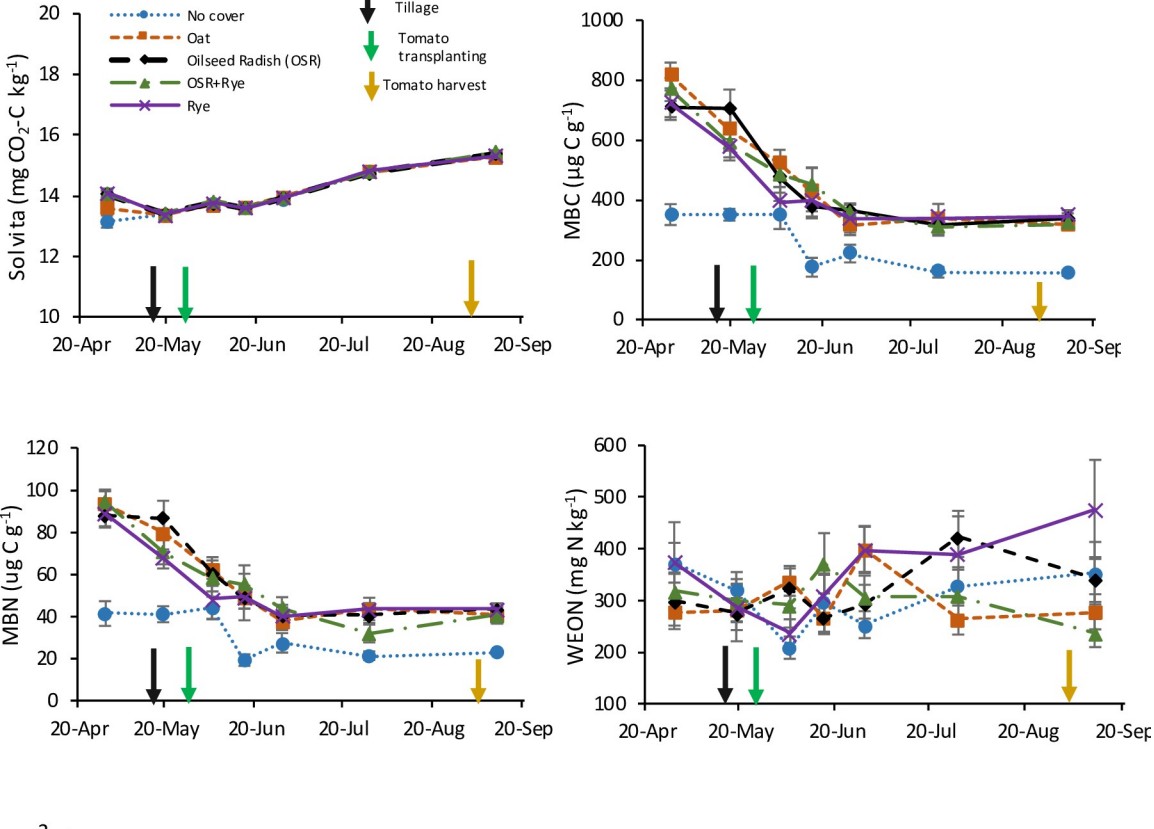

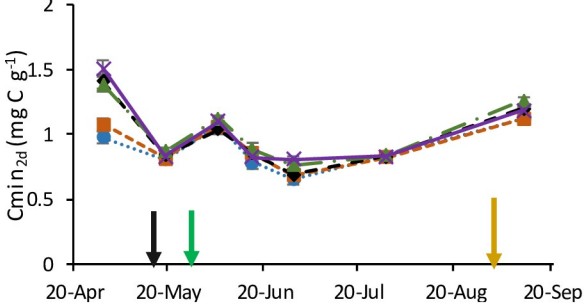

Sampling time

**Fig 3. In site-year 2015, effect of medium-term summer-planted cover crop (6-yrs) on soil labile fractions of C and N from 0–15 cm depth over the tomato growing season.** Bars represent standard error of means (n = 8). $Cmin_{2d}$, cumulative 2d soil C mineralization; MBC, microbial biomass C; MBN, microbial biomass N; WEON, water extractable organic N.

of soil attributes due to CC and crop residue treatments over the growing season confirms the dynamic nature of soil C and N fractions.

To facilitate comparison of CC and crop residue management on soil C and N fractions, radar charts were prepared for each site-year at tomato harvest (i.e., early September, Fig 6). In both site-years, even at one time-point (at tomato harvest representing cumulative effect of CC and crop residue treatments), differences in stable and labile fractions of C and N were detected with CC and crop residue treatments. Overall, in both site-years, the -R treatment had fewer differences among CC treatments than +R. No cover crop control had the lowest values of MBC, MBN, total N, OC, WAS, and $Cmin_{2d}$ in both crop residue treatments in site-

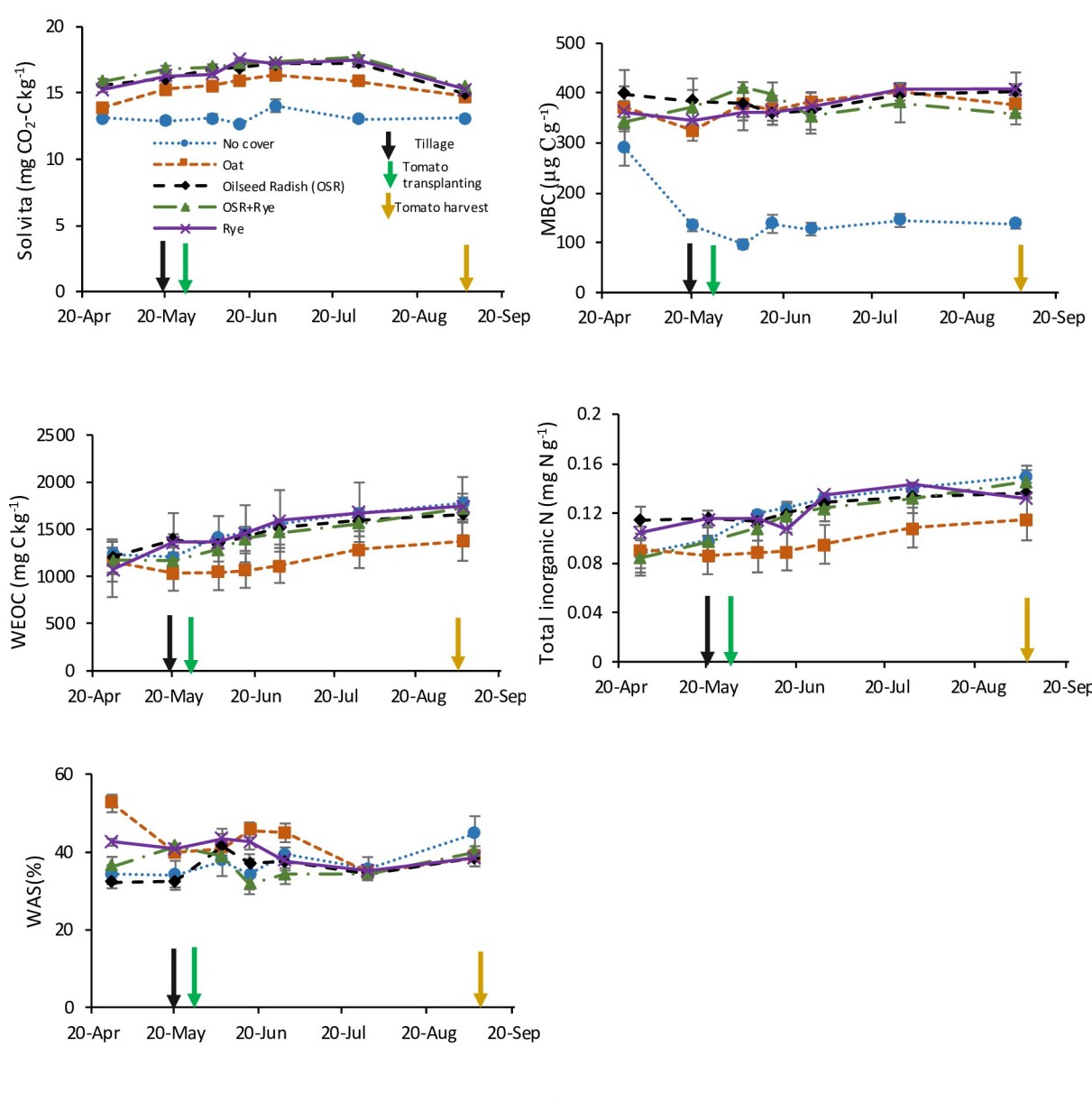

**Fig 4. Over the tomato growing season, effect of medium-term summer-planted cover crop (6-yrs) on soil labile fractions of C from 0–15 cm depth in site-year 2016.** Bars represent standard error of means (n = 8). MBC, microbial biomass C; WAS, wet aggregate stability; WEOC, water extractable organic C.

year 2015 (Fig 6). In addition to aforementioned fractions, no-CC was lowest for Solvita $CO_2$-burst in +R and -R in site-year 2016 (Fig 6). Overall between both site-years and crop residue treatments, cereal rye had the greatest values of $Cmin_{2d}$, MBC, MBN, OC, total N, and Solvita $CO_2$-burst. These CC effects (i.e, no-CC lowest MBC, MBN, total N, and OC but greatest SLAN) and crop residue effects on soil attributes at tomato harvest were consistent with treatment effects observed temporally over the crop growing season.

Across all the sampling times in site-year 2015 and 2016, correlation analysis was conducted between soil C and N fractions (S5 and S6 Tables). In site-year 2015, MBC correlated

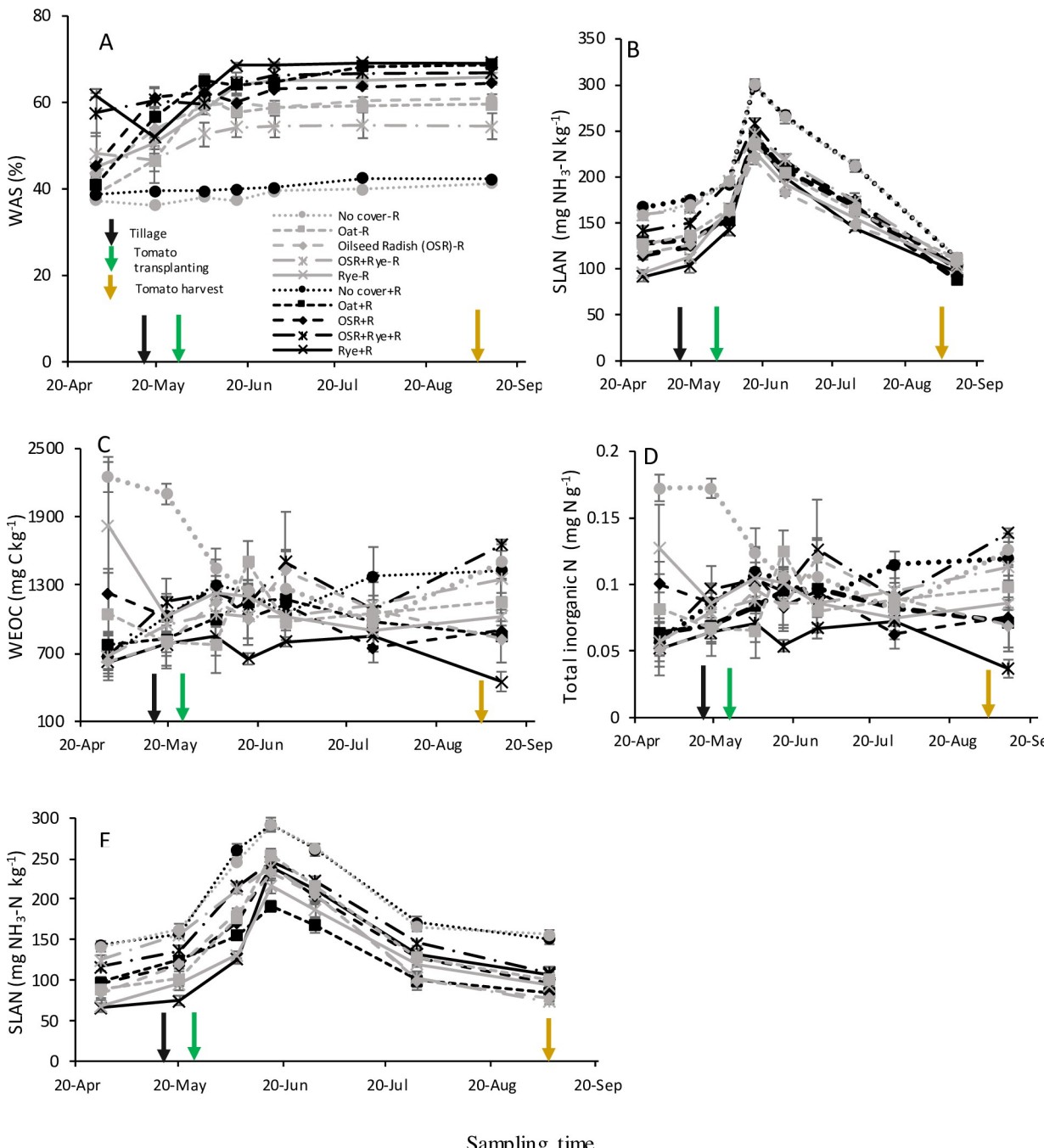

**Fig 5.** In site-year 2015 (A-D) and site-year 2016 (E), effect of medium-term summer-planted cover crop (6-yrs) and crop residue retained (+R) or removed (-R) on soil labile and stable fractions of C and N from 0–15 cm depth over the tomato growing season. Bars represent standard error of means (n = 4). SLAN, Solvita labile amino N; WAS, wet aggregate stability; WEOC, water extractable organic C; WEON, water extractable organic N.

negatively with Solvita $CO_2$-burst ($P < 0.0001$, r = -0.37), SLAN ($P < 0.001$, r = -0.34), total inorganic N ($P = 0.0004$; r = -0.21), and WEOC ($P = 0.0027$; r = -0.17) and positively with $Cmin_{2d}$ ($P < 0.0001$; r = 0.31). Similarly, in site-year 2015, MBN correlated negatively with Solvita $CO_2$-burst ($P < 0.0001$, r = -0.32), SLAN ($P < 0.001$, r = -0.35), total inorganic N

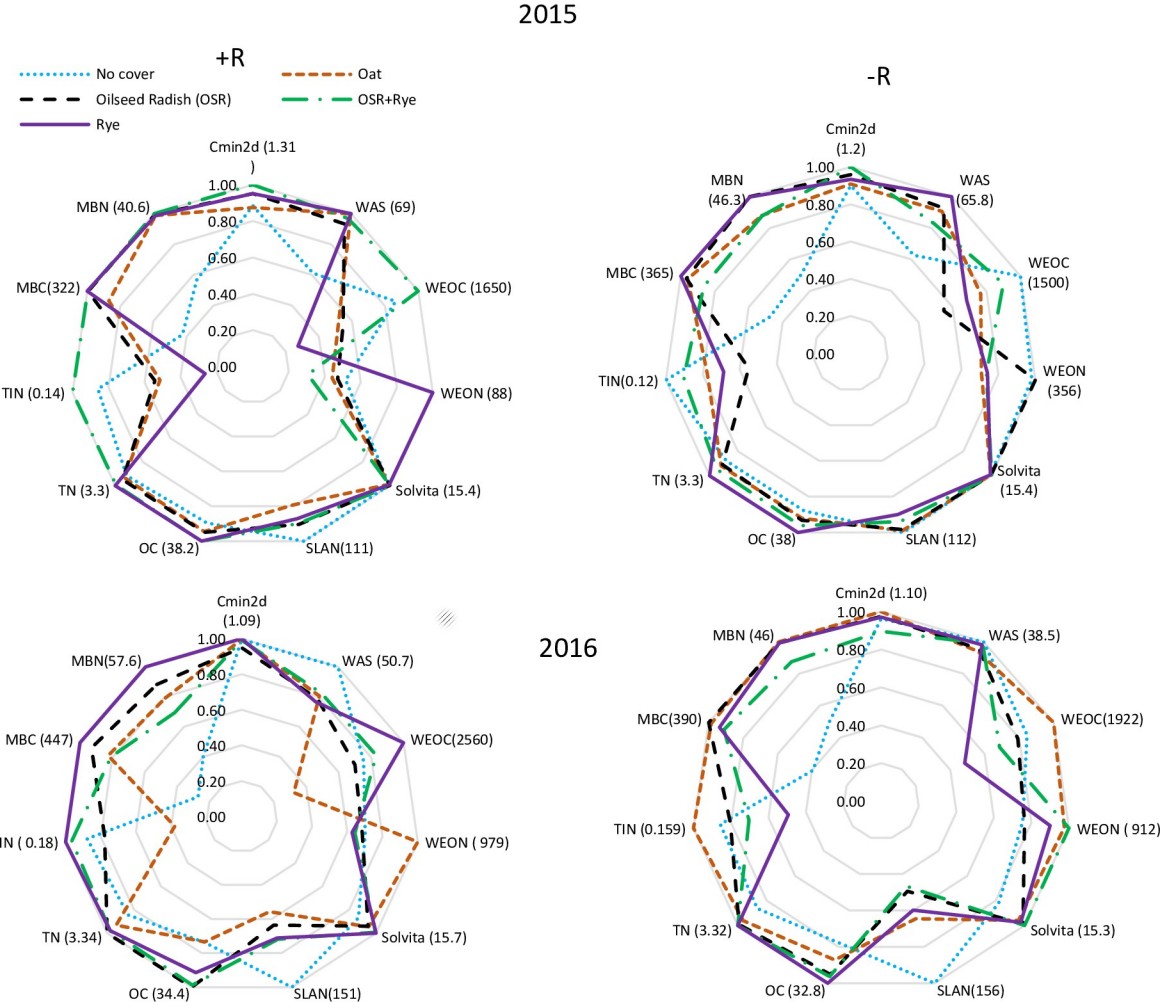

**Fig 6.** Effect of medium-term summer-planted cover crop (6-yrs) and crop residue retained (+R) (A,C) or removed (-R) (B,D) on soil labile and stable indicators of C and N at tomato harvest in site-year 2015 (A,B) and 2016 (C,D). Values in parenthesis represent mean (n = 4) maximum concentrations expressed as mg g$^{-1}$ for Cmin$_{2d}$, OC, TIN, and TN; mg kg$^{-1}$ for WEOC, WEON, Solvita, and SLAN; μg g$^{-1}$ for MBC and MBN; % for WAS (Cmin$_{2d}$, cumulative 2d soil C mineralization; MBC, microbial biomass C; MBN, microbial biomass N; SLAN, Solvita labile amino N; OC, organic C; TIN, total inorganic N; TN, total N; WAS, wet aggregate stability; WEOC, water extractable organic C; WEON, water extractable organic N).

($P$ = 0.0016; r = -0.18), and WEOC ($P$ = 0.0092; r = -0.15) and positively with Cmin$_{2d}$ ($P$ < 0.0001; r = 0.32).

Contrary to site-year 2015, in site-year 2016 there were fewer significant correlations (S6 Table). The only significant positive correlations were between WAS and Cmin$_{2d}$ ($P$ = 0.0189, r = 0.14), MBC and Solvita CO$_2$-burst ($P$ < 0.0001, r = 0.57), MBN and Cmin$_{2d}$ ($P$ = 0.0208, r = 0.13), and MBN and Solvita CO$_2$-burst ($P$ < 0.0001, r = 0.41). Moreover, correlations between OC-related processes (MBC, MBN, and WAS) and stable fractions (OC, total N) were assessed for both site-years (S5 and S6 Tables). Organic C correlated positively with WAS ($P$ < 0.0001, r = 0.24) in site-year 2015 but not in site-year 2016, whereas MBC ($P$ ≤ 0.0029, r = 0.17 to 0.57) and MBN ($P$ ≤ 0.0028, r = 0.17 to 0.41) correlated positively with total N in both site-years (S5 and S6 Tables). These correlations (positive or negative) between the short- and long-term indicators of management further confirm the close associations between the labile and stable fractions of C and N and their processes in our medium-term CC experiment.

## Discussion

In spring 2015 and 2016, CC C:N ratios were favourable for mineralization for all CCs (18.4 to 24.8) but OSR (32.3) in spring 2015 only. Similar C:N of CC was previously reported in this CC experiment [6,35] and elsewhere [11]. Our results of greatest CC biomass from OSR in fall 2014 and 2015 and OSR+Rye in spring 2015 and 2016 were consistent with the previous studies in the same medium-term CC experiment [6,34,35]. In this medium-term experiment, annual CC biomass production and C and N contents were different at each fall sampling (from fall 2007 to fall 2015 and from fall 2008 to fall 2016). This difference in fall CC biomass (and C and N content) over the study duration was due to timing of main crop harvested (i.e. fresh processing pea in late July and processing tomato in early Sept) and consequently the length of the CC growing season.

Cover crops are known to increase the concentrations of stable fractions of C and N in the long-term [53] but this effect is not observed in the short-term [54]. Moreover, several studies have demonstrated the strong influence of CC C and N inputs over the medium- and long-term in increasing the stable pools of soil C and N [53,55,56]. In our study, stable fractions of soil C and N (OC and total N) were influenced by CC and crop residue management but did not change temporally. We observed that, across all sampling times, incorporation of CC C and N inputs significantly increased the soil OC and total N concentration compared with the no-CC. A similar effect of CC treatments on soil OC and total N was reported by Zhou et al. [57]. Likewise, Sainju et al. [53,56] observed least total N concentration from no-CC. Lowest concentration of total N and OC in no-CC treatment could be attributed to lesser C and N biomass inputs than CCs in the medium-term. In our medium-term experiment, average annual CC above-ground C inputs were 1280 to 1380 kg C ha$^{-1}$ and CC above-ground N inputs were 78 to 97 kg N ha$^{-1}$, which contributed to a significant increase in soil OC and total N. Our observed greater OC and total N concentration with CCs and crop residue retention indicates the potential of these management practices to increase stable C and N in the medium-term and beyond.

As expected, labile fractions of soil C and N (WAS, WEOC, WEON, MBC, MBN, SLAN, Cmin$_{2d}$, Solvita CO$_2$-burst, and total organic N) changed with CC and crop residue treatments over the tomato growing season indicating the dynamic nature. Cover crop residue contributes to the labile soil organic matter, thus, affecting the C and N dynamics [58]. Labile pools of soil C and N are crucial for soil biology as they act as a nutrient reservoir for soil microbes [57]. Seasonal variation in the labile fractions observed might be attributed to the root exudates released from the tomato roots. As mentioned previously, tillage was conducted in mid May at both sites to incorporate the CC residues in soil and prepare the soil for tomato transplanting. Approximately a week after tillage, tomato seedlings were transplanted. Cover crop residues in addition to the root exudates and rhizodepositions released by the tomato roots undergo decay in soil over time and release labile compounds. Concentration of labile compounds is increased with an increase in plant density, size and development of root system. Therefore, greater concentration of rhizodeposits and labile fractions of C and N at the flowering than the planting stage is expected due to an increase in plant size and density at flowering (early to Mid-July in our production region). Several studies have reported that root C and N inputs are largely responsible for influencing the soil biological activity [59,60]. As the crop matures, the concentration of rhizo-deposits decreases but fresh crop residues and mature roots are added to the soil which contribute to the microbial substrates, therefore, increasing the labile fractions of C and N at harvest [61]. Overall, our observed variation in the labile C and N fractions across the growing season and between years might be attributed to the differences in the quantity of CC biomass produced among the tested CCs and the interactions between soil

microbes and plant roots. Lack of a clear distinction among the tested CC species in impacting the soil labile and stable C and N fractions and processes, such as respiration, across the season confirm the dominance of CC biomass (C inputs) in controlling the soil C and N dynamics than the quality (CC C:N) in this medium-term experiment.

Moreover, weather differences (air temperature and precipitation) between sampling years might have resulted in the temporal variation in the labile soil C and N. Similarly, seasonal differences in the labile fractions in response to variable frequency and intensity of precipitation were reported by Hui et al. [62]. Site-year 2016 was warmer in spring (April to May) and drier (from April to June) than 2015 (Fig 1), which likely impacted the microbial activity, residue decomposition, nutrient uptake, availability, and release. Generally, a direct and positive relationship is observed between temperature and microbial activity [63], while soil moisture content in the range of 50–70% of water holding capacity is preferred for microbial functions and processes [64]. Greater microbial activity results in an increase in the potential of residue decomposition. Our result of high MBC and MBN in early spring, perhaps due to low soil temperature and high soil moisture, was consistent with several studies [22–24]. Additionally, June 2015 had higher total precipitation than June 2016, which might have impacted the microbial mediated soil functions and processes between both site-years. Overall, high precipitation in July of both site-years was beneficial for the tomato crop as it matched with the period of greatest water demand by the crop.

Out of all the sampling times evaluated in this study, 2WAT (early June) and harvest (early September) is recommended for sampling soil C and N. In one out of two site-years, 2WAT had greater concentrations of Solvita $CO_2$-burst, WEON, $Cmin_{2d}$, and SLAN for all CC treatments than other sampling times, which might be attributed to an increase in the microbial activity and nutrient cycling due to residue incorporation. Tillage increases soil aeration and temperature, breaks soil macro and micro-aggregates; thus, exposes the physically protected soil OC [65] and increases the availability of OC to microbes. These processes result in increasing microbial activity and labile C and N parameters [66]. In agreement with our results, several studies [67–69] have reported that labile fractions of C and N have greater sensitivity to tillage than soil OC and total N; hence, have implications to be used as soil quality indicators. Moreover, early June (2WAT) represented the medium- (CC) and short- (residue incorporation) term effects on the tested indicators of soil quality. Likewise, Sainju et al. [7] reported increases in MBC, OC, and total N concentration following residue incorporation in a CC based tomato production system. Additionally, it has been observed that soil sampling for assessing N status should be conducted when the crop is actively growing and has an increased N demand. In our production system, tomato plants start actively growing and increasing in size in June, thereby suggesting an increase in demand and N uptake by tomato roots. Thus, June represents a suitable time for soil sampling to understand N dynamics in our production system. In contrast to June, microbial activity would be expected to be slower in April due to less availability of substrate and cool soil temperature, leading to the observed lower concentrations of WAS, SLAN, Solvita $CO_2$-C, total inorganic N, WEOC, and WEON. Hence, April may not be an optimum time for soil C and N sampling. Additionally, compared with other sampling times, greater treatment differences in soil labile fractions, such as WEOC, total inorganic N, Solvita $CO_2$-burst, SLAN, WAS, and MBC, were detected at tomato harvest. Results of larger pools of both stable and labile fractions with CC than no-CC at tomato harvest might be a reflection of the cumulative effects of CCing on soil build up and storage of C and N in our experiment. Therefore, June and/or September in a crop growing season is a reasonable time for soil sampling to evaluate soil C and N dynamics and overall soil quality. Furthermore, compared with the no-CC, CCs positively impacted the tomato primary productivity [33,36], further confirming the positive influences of CC on soil functionality and quality.

Compared to other CC treatments, lowest values of WAS, MBC, MBN, Solvita $CO_2$-burst, and $Cmin_{2d}$ were observed with no-CC (S3 and S4 Tables). A review by Blanco-Canqui et al. [11] reported greater percentage of water stable aggregates with CC than without. Soil aggregation is known to be impacted by the CC and crop residue retention due to an increased production of organic binding agents resulting in stabilization of aggregates [11,70]. Plant roots result in the formation of stable aggregates via the mechanisms of (a) increased production of polysaccharides, (b) trapping of soil particles between the root hairs, and (c) increased concentration of chemicals responsible for stabilizing micro and macroaggregates [71–73]. Similarly, several studies have reported the increases in MBC, MBN, and soil respiration (evolved $CO_2$) with CCs [74–77].

In only one of two site-years, -R lowered $Cmin_{2d}$, Solvita $CO_2$-burst, total inorganic N, and WEOC, which could be attributed to a reduction in microbial activity and C inputs with the crop residue removal [2]. Likewise, WAS was consistently lowest from -R plots in both site-years, which concurs with other research [5,13,70]. Inclusion of CC in cropping systems has shown to offset the negative effects of residue removal on soil C and N [78]; however, the extent to which CC mitigate the residue removal effects were highly dependent on CC biomass produced, duration of study, soil texture, initial OC concentration, and climatic conditions [2]. In contrast to removal, crop residue (winter wheat) retention (high C:N (80)) likely result in N immobilization, reduced crop N uptake, and decrease crop yield. But, the lack of any negative impacts of crop residue management on crop yield and C and N dynamics in our study might be attributed to inherently rich and fertile soil at our experimental sites.

Due to less differences observed in the CC parameters (quantity and quality of aboveground CC biomass) in our study, another mechanism (other than inputs) might be influencing the soil C and N pools and processes. Dignac et al. [79] also indicated the possibility of another abiotic OC stabilization mechanisms such as protection of OC within the minerals, adsorption and interaction of OC with the minerals, and physical protection of OC within the aggregates. Physical mechanisms of soil OC protect the degradation of OC by the microbes and help in the stabilization of soil OC in the long-term [80]. It is not clear if the observed increases in C and N attributes with CC and crop residue retention were due to minimizing C and N losses, protecting soil OC in micro-aggregates, preventing decomposition of organic matter, and/or influencing rhizodepositions and microbial community structures. Moreover, belowground biomass is another major contributor to soil OC concentrations, which was not measured in our study. Thus, future research elucidating the mechanism of CC (above- and below- ground biomass and root exudates) and crop residue effects on soil C and N cycling and storage is needed.

Additionally, the effects of land management practices on the mechanisms of soil C and N is not fully understood. Several meta-analyses and long-term studies have shown that the intensity of the mechanisms and processes affecting the dynamics and stabilization of soil C and N changes with time, land use, and climatic conditions [16,79–82]. Moreover, the processes governing the cycling of soil C and N are affected by the interactions with other nutrients present in the soil organic matter such as phosphorus, sulphur etc. [79]. Therefore, the aforementioned factors (temporal and environmental) should be considered while studying the processes and mechanisms influencing the soil C and N stabilization.

## Conclusions

Soil C and N fractions reflected both short- and medium-term dynamics in our study. Temporal variations in soil parameters were attributed to the short-term impact of CCs and crop residues on labile fractions of soil C and N (MBC, MBN, SLAN, Solvita $CO_2$-burst, WEOC,

WEON, total inorganic N, and Cmin$_{2d}$). The cumulative medium-term CC effect was greater concentrations of OC and total N (stable pools) with CC than without CC. Our observed differences (within and between site-years) in soil C and N fractions with CC and crop residue treatments might not be exclusive to the quantity and quality of CC biomass produced in this study, suggesting the possibility of another mechanism influencing soil C and N dynamics. Thus, future research evaluating the mechanism governing the pathway of soil C and N losses and gains with CC and crop residue management is needed.

Overall, positive influence of CCs and crop residue retention on soil C and N storage compared to no-CC and crop residue removal treatments indicates the impact of soil C inputs on microbial processes with implications on soil quality. This study improves our understanding of CC and crop residue removal effects on stable and labile fractions of C and N and indicates the potential of CC to offset partially the potential negative impacts of crop residue removal on soil quality in the medium-term. Additionally, our results demonstrated the role of CCs in increasing OC and total N concentration after 6-yrs of CCing and highlighted the potential of CCs in sequestering soil C and N with implications on mitigating climate change. Future research focusing on CC effects after crop residue removal with diverse land management practices, soil types, and cropping systems would enhance our understanding of the C and N cycling.

## Supporting information

**S1 Table. Impact of medium-term summer-planted cover crop (6-yrs) and crop residue management (retained (+R) or removed (-R) on soil moisture gravimetric content in site-years 2015 and 2016.** [a-c] In each column, based on a protected LSD test, means followed by a different letter indicate statistical significant difference ($P<0.05$). [z]SE, standard error of mean. (XLSX)

**S2 Table. Impact of medium-term summer-planted cover crops (6-yrs) (CC) on CC residue quality parameters (C concentration and C: N) during spring 2015 and 2016.** [a-b] For each parameter, based on a protected LSD test, means followed by a different letter indicate statistical significant difference ($P < 0.05$). [z]SE, standard error of mean. (XLSX)

**S3 Table. Impact of medium-term summer-planted cover crop (6-yrs) and crop residue retained (+R) or removed (-R) on soil labile and stable fractions of C and N from 0–15 cm depth in site-year 2015.** [a-d] For each fraction and effect, based on a protected LSD test, means followed by a different letter indicate statistical significant difference ($P<0.05$). Cmin$_{2d}$, cumulative 2d soil C mineralization; MBC, microbial biomass C; MBN, microbial biomass N; SLAN, Solvita labile amino N; OC, organic C;, total N; WAS, wet aggregate stability; WEOC, water extractable organic C; WEON, water extractable organic N. [z] SE, standard error of mean. (XLSX)

**S4 Table. Impact of medium-term summer-planted cover crop (6-yrs) and crop residue retained (+R) or removed (-R) on soil labile and stable fractions of C and N from 0–15 cm depth in site-year 2016.** [a-c] For each fraction and effect, based on a protected LSD test, means followed by a different letter indicate statistical significant difference ($P<0.05$). Cmin$_{2d}$, cumulative 2d soil C mineralization; MBC, microbial biomass C; MBN, microbial biomass N; SLAN, Solvita labile amino N; OC, organic C; WAS, wet aggregate stability; WEOC, water extractable organic C; WEON, water extractable organic N. [z] SE, standard error of mean. (XLSX)

**S5 Table. In a medium-term cover crop experiment, Spearman correlation coefficients between soil labile and stable fractions of C and N sampled from 0–15 cm depth in 2015.** *Significant correlation at $P < 0.05$. Cmin$_{2d}$, cumulative 2d soil carbon mineralization; MBC, microbial biomass C; MBN, microbial biomass N; SLAN, Solvita labile amino N; OC, soil organic C; WAS, wet aggregate stability; WEOC, water extractable organic C; WEON, water extractable organic N.
(XLSX)

**S6 Table. In a medium-term cover crop experiment, Spearman correlation coefficients between soil labile and stable fractions of C and N sampled from 0–15 cm depth in 2016.** *Significant correlation at $P < 0.05$. Cmin$_{2d}$, cumulative 2d soil carbon mineralization; MBC, microbial biomass C; MBN, microbial biomass N; SLAN, Solvita labile amino N; OC, soil organic C; WAS, wet aggregate stability; WEOC, water extractable organic C; WEON, water extractable organic N.
(XLSX)

## Acknowledgments

We appreciate the assistance of research technician, Mike Zink and co-op students for helping in the field and lab analysis.

## Author Contributions

**Conceptualization:** Laura L. Van Eerd.

**Formal analysis:** Inderjot Chahal.

**Funding acquisition:** Laura L. Van Eerd.

**Methodology:** Inderjot Chahal.

**Project administration:** Laura L. Van Eerd.

**Supervision:** Laura L. Van Eerd.

**Writing – original draft:** Inderjot Chahal.

**Writing – review & editing:** Inderjot Chahal, Laura L. Van Eerd.

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
