## [Decision Letter · Decision Letter 0]

12 Mar 2020

PONE-D-19-28936

Cover crop and crop residue removal effects on temporal dynamics of soil carbon and nitrogen in a temperate, humid climate

PLOS ONE

Dear Dr. Van Eerd,

Thank you for submitting your manuscript to PLOS ONE. After careful consideration, we feel that it has merit but does not fully meet PLOS ONE’s publication criteria as it currently stands. Therefore, we invite you to submit a revised version of the manuscript that addresses the points raised during the review process.

Dear authors, both reviewer recommended a rejection of the manuscript and made very detailed an in-topic comments. Both reviewers are experts in the field and I strongly take into account of their evaluation. I however only partly agree with them.

Plos 1 does not make assumptions on the novelty of the work and I think that the manuscript may have a chance.

Nonetheless, I completely agree with the reviewers’ concerns and I believe that all of them must be carefully addressed, along with some of mine below (in the section "Additional Editor Comments") to make the manuscript easily usable by a broad readership.

I thus invite you to take the revierwes’ comments in due account, and also adapt the writing style to a more general readership.

Regards.

We would appreciate receiving your revised manuscript by Apr 26 2020 11:59PM. To enhance the reproducibility of your results, we recommend that if applicable you deposit your laboratory protocols in protocols.io, where a protocol can be assigned its own identifier (DOI) such that it can be cited independently in the future. For instructions see: http://journals.plos.org/plosone/s/submission-guidelines#loc-laboratory-protocols

We look forward to receiving your revised manuscript.

Kind regards,

Sergio Saia, Ph.D.

Academic Editor

PLOS ONE

Additional Editor Comments:

RECOMMENDATION:

Dear authors, both reviewer recommended a rejection of the manuscript and made very detailed an in-topic comments. Both reviewers are experts in the field and I strongly take into account of their evaluation. I however only partly agree with them.

Plos 1 does not make assumptions on the novelty of the work and I think that the manuscript may have a chance.

Nonetheless, I completely agree with the reviewers’ concerns and I believe that all of them must be carefully addressed, along with some of mine below, to make the manuscript easily usable by a broad readership.

I thus invite you to take the revierwes’ comments in due account, and also adapt the writing style to a more general readership.

Regards.

ABSTRACT

Please note that I revised the abstract before reading the whole manuscript. Additional comments on the abstract after reading the manuscript, if any, can be found at the end of this revision (Additional suggestions)

The abstract is a little bit puzzling. The reader has no clear idea of the setup, and you use a writing style that leave too much aspects on the reader’s perception. I suggest to better describe the setup, especially in the methodological aspects.

Just a suggestion as a non-native speaker. Non-native speakers have problems in perceiving the correct meaning of “Temporally”. I suggest to change it.

INTRODUCTION

L42-44: unclear how and why

L85: please better link it to the previous sentences

L86-90: the aim as it is described is quite confusing. The reader has no idea of the implication of the years, nor it’s needed here. Just say the aim with few, but important, methodological aspects.

Were CC and residues management in interaction or not? What does residue management means? If you mean “removal or not”, just declare it. How many tomato growing seasons were studied?

L90: with regards to the stabile, this hypothesis contrast with the hypothesis 2 unless you studied a long term, too. And this is unclear.

MATERIALS AND METHODS

L94: quiet unclear, the reader mustn’t be obliged to read the others, just recall those setups.

L95-98: I think this could be efficiently reported in a time-picture

L166: protected LSD is marginally ok for your setup. Since the data are balanced, you may have had few or no problems of difference estimation. For your further work, I suggest to use pdifferences of the LSmeans, eventually corrected by Tukey Kramer depending on the balancing of the setup.

RESULTS

L202: as an editor, I am not interested in the shape you give to the figure, but I suggest to ass year as a random factor or, if retaining this analysis, split the years in the figures, with one year in a side and the other in the other side. Treatments are within years, not its contrary. The results of the analysis you are providing are unclear. The reader wants to know the mean effect of the treatments across years. Lines recall a trend, but this is not the case.

L222: providing the p<0.05 in bold helps in reading

L250: this pc must be transformed in order to be more readable. Please change colors, sizes, and when applicable y-axes. Do the same for fig. 4

DISCUSSION and CONCLUSIONS

L417-418: I suggest to remove this sentence

Journal Requirements:

Reviewers' comments:

Reviewer's Responses to Questions

**Comments to the Author**

1. Is the manuscript technically sound, and do the data support the conclusions?

Reviewer #1: Yes

Reviewer #2: Partly

2. Has the statistical analysis been performed appropriately and rigorously? 

Reviewer #1: Yes

Reviewer #2: No

3. Have the authors made all data underlying the findings in their manuscript fully available?

Reviewer #1: Yes

Reviewer #2: No

4. Is the manuscript presented in an intelligible fashion and written in standard English?

Reviewer #1: No

Reviewer #2: No

5. Review Comments to the Author

Reviewer #1: This manuscript reports the results of a study aimed at evaluating the short and long term effect on soil C and N indicators of cover crops and crop residues incorporation in the soil. The paper is clearly written and all the sections are appropriately developed. Table are well designed and clear. Unfortunately, the quality of the figure is not adequate at all: they are totally fuzzy, the resolution is too low and they are unintelligible.

According to my opinion the results about the so defined “short term” indicators are poorly discussed in the light of the interaction of the main experimental factors with the crop (tomato) and its growing phases, which may have influenced the responses. In this regards, I would suggest the Authors to analyse/plot the indicators results versus the crop phenological development stages (i.e. BBCH-scale) instead of time (weeks). This approach could help to understand the potential effect of crop growing and its effect, via rhizosphere equilibrium, on C and N short term soil indicators.

However, the main flak of the paper is that, in my opinion, it is not novel enough. The most of the considerations and the conclusions are already known and no new information is reported in the manuscript. For this reason I would not suggest to accept the paper for publication in Plos One.

Reviewer #2: Review PONE-D-19-28936

Cover crop and crop residue removal effects on temporal dynamics of soil carbon and nitrogen in a temperate, humid climate

by Chahal & Van Eerd

Content

The paper describes the impact of 4 cover crops planted 6 years and the use of crop residues 3 times in a field experiment on soil quality indicators associated with the C and N cycles. The field trial was conducted twice, starting in 2007 and 2008. Apart from cumulative effects of cover crops and residue management on soil quality indicators across the live span of the field experiment, the time course in the growth season of tomatoes was measured my multiple sampling to assess short term effects of cover crops.

The authors found, that both, cover crops and crop residues increased soil organic carbon and total nitrogen. As expected, there were no seasonal changes of soil organic carbon and total nitrogen, but label carbon and nitrogen fractions, soil respiration and microbial biomass N and C were affected.

Although the findings are interesting, I have several concerns how the manuscript is presented and written.

General comments

1. The language and writing style is very technical, using a lot of abbreviations. The way how the manuscript is written, it is more addressing a specialist readership, and to a less extent a general audience. In addition, the text is often not clear, for example, the abstract is not understandable without reading the full paper. What are time-years, double parenthesis are not easy to read, etc.

2. The text contains some un-logic constructions with use of several terms not in a systematic manner.

Expl. 1: line 83-84: “The effect of soil functions and treatments on seasonal (or short-term) trends and microbial processes suggests the need for short-term assessments of soil C to avoid misinterpretation of treatment effects on the C dynamics.” Soil functions do not effect seasonal trends and microbial processes. Soil functions, eg N cycling is the result of microbial processes and is affected by cover crops.

Expl. 2: line 427-429: “Overall, compared to no-CC and crop residue removal treatments, CCs and crop residue retention had greater concentrations of soil parameters temporally indicating the impact on soil C inputs and microbial processes have implications on soil quality”. It may read “… the impact of soil C inputs on microbial processes with implications on soil quality”.

3. The hypothesis is not original. “We hypothesize that (i) short-term (seasonal) changes will be detected in the labile indicators of soil C and N across the tomato growing season, (ii) no seasonal variability will be detected in the stable indicators of soil C and N, and (iii) compared with the no cover crop control (no-CC) and crop residue removal (-R), CCs and crop residue retention (+R) treatments will have greater concentrations of labile and stable indicators of soil C and N”. We expect that so-called stable indicators like soil organic carbon and total nitrogen in the soil does not change within season, but label fractions are affected. This is textbook knowledge. I suggest focusing on the question, if cover crops and crop residues can maintain soil organic carbon and total nitrogen, and if there are additive effects. Interactions of labile indicators within season should be discussed in view of their ecological and agronomic relevance, correlations are not so meaningful. To conclude that labile fractions are reflecting seasonal changes is trivial.

PS: Instead of labile and stable indicators, I would prefer to refer to indicators for more stable soil parameters, and indicators for more labile soil quality parameters. At least define what are stable and labile indicators.

4. The claim that sampling 2 weaks after tillage and after harvest are optimum sampling time points needs further statistical analyses, using multivariate statistics.

In all, I cannot recommend the manuscript for publication in PLOS ONE as it stands.

Detailed comments

L 61: Microbial activity is not a soil process

L107: The field trial should be described briefly

L117: Were biomass N concentrations also determined?

L120: What is a site-year?

L121: Was tillage connected to the incorporation of the cover crops?

L168: What is a mean maximum value?

L173: Growing conditions: were tomato plants irrigated? Should it read growth season?

L258ff: You may only refer to differences, which are significant

L298-L315: Extremely long text just to describe correlations. I suggest to include a table in the supplement, and to show most important findings as graph in the main part.

L322: Difference in CC biomass and eventually C and N content. Probably? Reasoning?

L334: N inputs were 78 to 97 kg ha-1. This is N taken up from soil, no new N inputs

L345: Is the method to measure respiration precise enough to discern between cover crops? Did you once apply a standard method for comparison?

L 359ff: To justify that sampling 2 weeks after tillage and after harvest is recommendable to discern short- and medium-term effects of cover crops, you may apply multivariate statistics.

L378: What do you mean by: “Peak period of tomato growth initiates in June and greater

indicator concentration during June might have increased the nutrient availability to tomato roots and hence increased crop yield». What is the meaning of this sentence? You did not present yields, and these relations are extremely speculative.

L387: You did not find a increase of respiration by cover crops.

L407: Root exudates of CC may also be mentioned to positively affect stable and labile soil quality indicators.

L420: TN is not a short-term indicator

L427-L429: Far from clear

L433ff: These conclusions are not based on your results, but rather general.

Tab2 and 3: I suggest to put these tables in the annex

6. PLOS authors have the option to publish the peer review history of their article (what does this mean?). If published, this will include your full peer review and any attached files.

Reviewer #1: No

Reviewer #2: No

---

## [Author Response · Author response to Decision Letter 0]

22 Apr 2020

Thank you for your comments and suggestions. We have responded and addressed all the reviewer and editor comments in the manuscript. We have attached the "response to reviewers" file along with revised manuscript submission.

---

## [Editor Report · Decision Letter 1]

11 May 2020

PONE-D-19-28936R1

Cover crop and crop residue removal effects on temporal dynamics of soil carbon and nitrogen in a temperate, humid climate

PLOS ONE

Dear Dr. Van Eerd,

Thank you for submitting your manuscript to PLOS ONE. After careful consideration, we feel that it has merit but does not fully meet PLOS ONE’s publication criteria as it currently stands. Therefore, we invite you to submit a revised version of the manuscript that addresses the points raised during the review process.

Please see the the "Additional Editor Comments" section for further indications.

We would appreciate receiving your revised manuscript by Jun 25 2020 11:59PM. To enhance the reproducibility of your results, we recommend that if applicable you deposit your laboratory protocols in protocols.io, where a protocol can be assigned its own identifier (DOI) such that it can be cited independently in the future. For instructions see: http://journals.plos.org/plosone/s/submission-guidelines#loc-laboratory-protocols

We look forward to receiving your revised manuscript.

Kind regards,

Sergio Saia, Ph.D.

Academic Editor

PLOS ONE

Journal Requirements:

see below

Additional Editor Comments:

Dear authors,

I think that you answered the suggestions by the reviewers and your manuscript definitely improved.

I passed over the reviewers opinion on the novelty because of the Plos1 policy, which <<evaluate submitted manuscripts on the basis of methodological rigor and high ethical standards, regardless of perceived novelty>> [***https://journals.plos.org/plosone/s/journal-information#loc-scope***].

I uploaded you a tracked change version with few comments to address before the manuscript can be judged acceptable.

In addition, I must point that << PLOS journals require authors to make all data underlying the findings described in their manuscript fully available without restriction, with rare exception.>> [***https://journals.plos.org/plosone/s/submission-guidelines#loc-materials-and-methods*** , section “*Data*” and also ***https://journals.plos.org/plosone/s/data-availability***]. Please take a look at these links.

I think your data should not represent a large data set and CSV or excel files could be suitable to upload it in the supplementary material (as also indicated in the link above)unless you uploaded it in other repositories or have some legal limitation to upload it.

In order to upload data, feel free to take as an example the dataset I uploaded in recent work of mine in the supplementary material of Saia et al. Mycorrhiza 2020, DOI: 10.1007/s00572-019-00927-w.

Please take into account that this is not an invitation to cite my article, which has nothing to do with your manuscript, and is only limited to show you the data needed to upload. In particular, the information on the variability of the data (separately per year and treatment) you presented in the manuscript if fundamental to both meet the plos1 policy and allow the inclusion of the data in further analyses (e.g. meta-analyses).

Regards

Sergio Saia

---

## [Author Response · Author response to Decision Letter 1]

15 Jun 2020

All editor/reviewer commends responded to in Response file attached.

---

## [Editor Report · Decision Letter 2]

22 Jun 2020

Cover crop and crop residue removal effects on temporal dynamics of soil carbon and nitrogen in a temperate, humid climate

PONE-D-19-28936R2

Dear Dr. Van Eerd,

We’re pleased to inform you that your manuscript has been judged scientifically suitable for publication and will be formally accepted for publication once it meets all outstanding technical requirements.

Kind regards,

Sergio Saia, Ph.D.

Academic Editor

PLOS ONE
---

## [Editor Report · Acceptance letter]

29 Jun 2020

PONE-D-19-28936R2 

Cover crop and crop residue removal effects on temporal dynamics of soil carbon and nitrogen in a temperate, humid climate 

Dear Dr. Van Eerd:

I'm pleased to inform you that your manuscript has been deemed suitable for publication in PLOS ONE. Congratulations! Your manuscript is now with our production department. 

Kind regards, 

on behalf of

Dr. Sergio Saia 

Academic Editor

PLOS ONE